# Update on Coagulase-Negative Staphylococci—What the Clinician Should Know

**DOI:** 10.3390/microorganisms9040830

**Published:** 2021-04-14

**Authors:** Ricarda Michels, Katharina Last, Sören L. Becker, Cihan Papan

**Affiliations:** Institute of Medical Microbiology and Hygiene, Saarland University, 66424 Homburg, Germany; ricarda-michels@web.de (R.M.); katharina.last@uks.eu (K.L.); soeren.becker@uks.eu (S.L.B.)

**Keywords:** coagulase-negative staphylococci, hospital-acquired infections, foreign body-related infections

## Abstract

Coagulase-negative staphylococci (CoNS) are among the most frequently recovered bacteria in routine clinical care. Their incidence has steadily increased over the past decades in parallel to the advancement in medicine, especially in regard to the utilization of foreign body devices. Many new species have been described within the past years, while clinical information to most of those species is still sparse. In addition, interspecies differences that render some species more virulent than others have to be taken into account. The distinct populations in which CoNS infections play a prominent role are preterm neonates, patients with implanted medical devices, immunodeficient patients, and those with other relevant comorbidities. Due to the property of CoNS to colonize the human skin, contamination of blood cultures or other samples occurs frequently. Hence, the main diagnostic hurdle is to correctly identify the cases in which CoNS are causative agents rather than contaminants. However, neither phenotypic nor genetic tools have been able to provide a satisfying solution to this problem. Another dilemma of CoNS in clinical practice pertains to their extensive antimicrobial resistance profile, especially in healthcare settings. Therefore, true infections caused by CoNS most often necessitate the use of second-line antimicrobial drugs.

## 1. Introduction

Coagulase-negative staphylococci (CoNS) form a large group of Gram-positive cocci united by their mutual lack of the virulence factor coagulase [1]. Many species belong to this group, the latest one, *Staphylococcus borealis*, being described as recent as 2020 [2]. In daily clinical practice, CoNS are commonly regarded as less pathogenic than *Staphylococcus* (*S*.) *aureus* and other members of the *S*. *aureus* complex [3,4,5], which in contrast possess coagulase. There is a huge discrepancy in published literature, with publications pertinent to *S*. *aureus* outnumbering literature on CoNS by far. This discrepancy is indicative of the relative lack of CoNS’ scientific appraisal during the past years and decades. Many CoNS infections are associated with foreign bodies (e.g., catheters) that facilitate biofilm formation, which contributes to CoNS pathogenicity. Importantly, the virulence factors of CoNS vary considerably, with some species (e.g., *S*. *lugdunensis*) being capable of significant adverse impacts on patients [6,7,8]. A probable explanation to why CoNS-related infections are overlooked so often is the fact that they are frequent commensal members of the skin microbiota [9]. As a consequence, they are often classified as contaminants rather than the causative agent of infection. The distinction between infection and contamination is not always straightforward, and most attempts thus far to identify one or more distinctive marker(s) have been unsuccessful. In this review, we will focus on the increasing clinical impact of CoNS-associated infections, the challenges in diagnostics, and the current therapeutic options.

## 2. Increasing Clinical Impact of CoNS

CoNS have been increasingly recovered in clinically relevant samples, e.g., blood cultures or otherwise primarily sterile samples, in parallel to the advancement in all medical specialties [10]. These medical advancements include not only sophisticated immunosuppressive or -modulatory treatment regimens in oncology, but also the increased use of implantable foreign bodies, such as central venous access devices, total joint replacements, and vascular grafts. 

Newborns and preterm neonates are a particularly vulnerable patient group, in whom infections caused by microorganisms with low pathogenicity can occur at a higher rate than in otherwise healthy children or adults [11,12,13]. Neonatal sepsis contributes heavily to the high morbidity and mortality, and CoNS have been reported to be one of the main causes of neonatal sepsis in neonatal intensive care units (NICU) [14]. However, bloodstream infections (BSI) caused by CoNS are one of the most prevalent nosocomial infections among all age groups. CoNS represented 31% among all cases of nosocomial BSI within a period of 7 years in a total of 49 US hospitals [15]. This finding was confirmed in several other cohorts of different geographical backgrounds over the past years [16]. For example, an observational study from Germany, which examined the prevalence of nosocomial infections in a University hospital, identified CoNS as the second most common cause of nosocomial infections [17]. More recent studies revealed the relationship between CoNS and the increased use of implanted medical devices like cardiac valves or joint replacements [18]. 

Several CoNS form a biofilm that enables the bacteria to adhere to medical devices and, as a result, to protect them against antibiotics [19]. Furthermore, especially elderly people with significant comorbidities, premature neonates or immunocompromised patients are at high risk for CoNS-associated BSIs, skin and soft tissue infections, and both native and prosthetic valve endocarditis. Additionally, CoNS-associated infections can occur in young and healthy individuals as well. *S. saprophyticus*, for example, is a cause for urinary tract infections, especially in young women. 

Moreover, CoNS are being increasingly studied within veterinary medicine, while their role as disease-causing pathogens in animals is still regarded as small. Although the relevance for human disease is not fully established, it has been shown that CoNS-inhabiting animals can display a wide range of antimicrobial resistances, and thus may potentially serve as a reservoir of resistance genes [20]. 

Currently, more than 50 different CoNS species have been described. Figure 1 displays an overview of CoNS known thus far, their attributable clinical relevance, frequency, and the associated infectious syndrome. In total, six species are believed to be associated with a higher clinical significance, namely *S. epidermidis, S. saprophyticus, S*. *haemolyticus*, *S. capitis*, *S. hominis,* and *S. lugdunensis*. Some species, including *S. lugdunensis*, are known to cause severe clinical disease. Antibiotic resistance has become an increasing problem with *S. lugdunensis*, which has some mutual features with *S. aureus* and has been reported as a cause of infective endocarditis [6,8].

Besides sporadic case reports, only limited clinical data is available about the most recently discovered CoNS species. Six species were discovered between 2015–2020, one of which is *S. argensis* [22]. Up until now, this strain has only been isolated from an aquatic environment, precisely from the river, Argen, in Southern Germany. *S. edaphicus* was also isolated from a natural environment. Scientists first isolated *S*. *edaphicus* from stone fragments and sandy soil in James Ross Island, Antarctica [23]. It is notable that *S. edaphicus* possesses mobile genetic elements that carry antimicrobial resistance genes, thereby rendering it a resistance reservoir that potentially spreads resistance-associated genes. Three further species were described for the first time in 2019, i.e., *S. caeli*, *S. pseudoxylosus,* and *S. debuckii* [24,25,26]. All three were isolated from various animal environments (*S. caeli* from air sampling in a rabbit holding, *S. pseudoxylosus* from bovine mastitis and *S. debuckii* from bovine milk), and their clinical relevance for humans remains to be established. The most recently discovered species, *S. borealis*, was obtained from four isolates of the human skin, as well as from one blood culture, indicating this species’ ability to colonize human skin and potentially penetrate into the blood stream [2]. Due to lack of data, it has not been possible yet to link the newer CoNS species to specific infectious syndromes or define their impacts on humans. However, one existing problem is that even species that have been known for a longer time can be difficult to assign to a specific infectious syndrome. In order to give an exemplary overview into a selection of CoNS and the wide range of associated infections, three species and their characteristics of clinical cases reported thus far are listed in Table 1. For *S. saccharolyticus*—of note, the only anaerobic CoNS—more than a dozen clinical cases have been reported. Several of these cases were foreign body-related infections, but also cases without any indwelling medical devices have been documented [27,28,29,30,31,32,33,34,35,36,37,38,39,40,41,42]. Only single case reports for the more recently described species *S. massiliensis*, *S. petrasii subsp. petrasii*, and *S. petrasii subsp. croceilyticus* are existing [43,44]. However, this does not necessarily translate into a reduced clinical significance. It was pointed out by the authors of the case report that a variety of CoNS species were found in the clinical samples, which made it impossible to undoubtedly identify the species that caused the infection and thus determine the clinical significance of a species. This has been a recurring problem in the diagnosis of CoNS. 

## 3. The Diagnostic Complexities—How to Distinguish between Infection and Contamination

The main challenge in diagnosis is the correct adjudication of whether or not the detected organism, i.e., a CoNS member, is the causative pathogen of the patient’s infection. It can be assumed that the complexity of a diagnosis leads to potentially disregarded CoNS infections, consequently leading to undertreatment (i.e., delayed or withheld antibiotics) in some cases and thereby contributing to morbidity and mortality [45]. On the other hand, antibiotic overtreatment is associated with the development of antimicrobial resistance, severe adverse events, and higher costs, mainly due to the need for in-patient therapy [46]. Ultimately, an unnecessarily prolonged hospitalization does not only cause a loss in quality-adjusted life-years, but hospital bed occupancy exposes another factor of this complexity, which becomes particularly immanent during times of hospital capacity shortage, especially during pandemics, such as COVID-19 [47]. 

Another diagnostic challenge is the correct identification on a species level. In some laboratories, matrix-assisted laser desorption/ionization time of flight mass spectrometry (MALDI-TOF MS) may not be as readily available as in others, even though this method has been considered the gold standard since its introduction into microbiological diagnostics more than a decade ago [48,49,50]. As described above, significant interspecies differences in clinical relevance, pathogenicity, and even antimicrobial susceptibility exist [50,51]. Therefore, identification to the species level can be regarded as fundamental. Yet, taxonomic categorizations are ongoing. Some previously described species have been partially re-classified, and some newly described CoNS may not yet be deposited in the relevant MALDI-TOF MS databases [52]. An additional problem that can complicate the microbiological identification is the fastidious and slow growth of some species, for example of *S. saccharolyticus* [38]. 

The group of CoNS is a heterogenous group constituted by a variety of different species, each with a unique set of traits. These microbiological properties have previously been described in detail [53]. Nevertheless, many studies have the tendency to compile all CoNS into one group. Not only do the phenotypic traits differ, but the niches which they tend to colonize vary widely. For example, *S. capitis* is found preferably on the scalp, while *S. cohnii* is *isolated* from feet, and *S. saccharolyticus* from back skin [40,54]. Considering the CoNS trait to colonize the human skin as commensals, it is not surprising that they can result in contaminated samples. Currently it remains a challenge to find valid and universally applicable tools that make it possible to decide whether the clinical case is an infection caused by CoNS or if the detection of a CoNS species indicates just a contamination with the same respective microorganism. Figure 2 summarizes a proposed set of criteria for the distinction between contamination and infection derived from the literature. 

Beekmann and colleagues proposed that in order to prevent misclassification of CoNS as a cause of infection, at least two independent blood cultures must be positive for CoNS within 5 days. Alternatively, one positive blood culture suffices if the clinical symptoms are suggestive of an infection [55]. Other research groups have similarly reasoned that one single positive blood culture may be clinically significant, especially in the context of clinical syndromes with high mortality, e.g., sepsis [56]. García-Vázquez proposed an algorithm including a Charlson score ≥ 3, Pitt score ≥ 1, neutropenia, the presence of central venous catheter, identification of *S*. *epidermidis*, and time to positivity < 16 h, which yielded a positive predictive value of 83% [57]. Apart from this, additional criteria can be considered. For example, Hitzenbichler and colleagues performed a single-center retrospective analysis of 252 patients with blood cultures positive for CoNS other than *S*. *epidermidis* [54]. They considered an infection as “likely” when all of the following criteria where met: absence of another likely infection at the time of blood culture withdrawal; ≥ 2 blood cultures were positive with the same species or one of the findings was a relevant clinical specimen; and if the symptoms or markers of inflammation improved after therapy. To account for cases with only one positive blood culture, they designated these as “possible” infections, if in addition a foreign body was in situ. Additionally, according to the authors, the time to positivity, the resistance profile, and the sole growth in anaerobic blood cultures should also be examined for their potential do distinguish between infection and contamination. Leveraging sequencing or genotyping data to characterize the pathogenic potential of CoNS constitutes another approach that has been applied recently [58]. Aiming to find a genotype-phenotype correlation, Shelburne and colleagues showed with whole-genome sequencing that certain sequence types of *S*. *epidermidis* predominate among patients with complicated BSI [59]. Sánchez and colleagues sought to find a genetic marker to distinguish between commensal and infecting strains of *S. epidermidis* in prosthetic joint infections [60]. While some differences between the strains were noted, especially pertaining to antibiotic resistance and genes linked with increased pathogenicity, properties like biofilm formation were equally distributed. They concluded that despite some differences, no single distinction marker, which would be sensitive and specific enough, could be found. 

In conclusion, it can be summarized that several distinguishing features exist, but a general and reliable method remains elusive. Another, yet different approach would entail taking the host response into account. This can be achieved by either measuring novel biomarkers or sets thereof [61], or by exploiting transcriptomics to measure differential gene expression [62]. Several research groups have developed transcriptomic tools utilizing different gene sets, while the feasibility on a large-scale and in point-of-care settings remains to be established [63,64]. We hypothesize that future steps could combine both modern microbiological methods with probing the host response. However, clinical data employing this dual approach are currently very scarce [65,66]. 

**Table 1 microorganisms-09-00830-t001:** Detailed clinical information on the cases reported thus far with *S*. *saccharolyticus*, *S*. *massiliensis*, and the different *S*. *petrasii* subspecies.

Species	Main Source	Case Reports	References
*S. saccharolyticus*	Human skin (especially back skin), animal skin (gorilla), contaminated platelet concentrates	13 Anaerobic endocarditis, prosthetic valve endocarditis, bacteremia, discitis and vertebral osteomyelitis, pneumonia, lung infections in cystic fibrosis patients, infection of the shoulder joint, bone marrow infection, pyomyositis, heart valve disease, spondylodiscitis, empyema.	[27,28,29,30,31,32,33,35,36,37,39,41,42]
*S. massiliensis*	Human skin	1 Brain abscess.	[43,67,68]
*S. petrasii*			
subsp. *petrasii*	Human skin and ear canal	1 Cerebral hemorrhage.	[44]
subsp. *croceilyticus*	Human skin and ear canal	1 Acute otitis externa.	[44]
subsp. *jettensis*	not documented	/ Strains were isolated from human clinical samples which were expected to be sterile (catheters, biopsies, cerebrospinal fluid, blood and deep swabs). Moreover, they were found in mixtures with other CoNS, which made it difficult to assess their clinical significance.	[69,70]
subsp. *pragensis*	not documented	6 Prostatitis, hand wound infection, appendicitis, pancreatitis, sepsis, phlegmona.	[71]

## 4. Therapy and Prevention—Many Old Bugs, Little New Drugs

### 4.1. General Remarks

As opposed to strains acquired in community settings, nosocomial or healthcare-associated CoNS usually display a wider range of resistance patterns. Most notable are the higher resistance rates to beta-lactam antibiotics, including penicillin, oxacillin/methicillin, but also gentamicin, clindamycin, ciprofloxacin, and erythromycin. Hence, CoNS can be considered as relatively “difficult to treat”, which is somewhat in stark contrast to the aforementioned low pathogenicity compared to *S*. *aureus* or others. Moreover, CoNS have the ability to rapidly acquire and modify resistance genes. This ability subsequently promotes the transmission of these genes into different staphylococcal species or even other bacterial genera [72,73]. While methicillin-susceptible strains can and should be treated like methicillin-susceptible *S. aureus* infections, i.e., with an anti-staphylococcal penicillin e.g., (flucl-)oxacillin (or alternatively, cefazolin), the majority of CoNS in health-care settings require the use of second-line antibiotics such as vancomycin, daptomycin, or linezolid. Specific therapy recommendations depend on the affected organ system; on the presence of foreign bodies, the contingency to remove those; and patient characteristics, such as age, immune status, and comorbidities. 

### 4.2. Glycopeptides

Vancomycin, the oldest glycopeptide antibiotic, offers a broad Gram-positive coverage through inhibiting cell wall synthesis [74]. Initially, its use was propagated for the treatment of penicillin-resistant *Staphylococcus aureus*, but with the advent of penicillinase-stable beta-lactam antibiotics, it soon became a reserve substance, e.g., for patients allergic to beta-lactams, and the drug of choice for infections caused by methicillin-resistant *S. aureus* (MRSA). Although the increasing prevalence of vancomycin-resistant enterococci (VRE) has limited its empiric use in settings with high VRE prevalence, the rate of vancomycin-resistance among staphylococci has remained steadily low [75]. Therapeutic drug monitoring (TDM) is recommended to ensure target attainment and to minimize drug toxicity, mainly nephrotoxicity. The so-called “red man syndrome” has been associated with rapid infusion. A recently revised guideline of the Infectious Disease Society of America (IDSA) has proposed a more sophisticated approach to TDM in the context of MRSA. It propagates computing the target area under the curve over 24 h to minimum inhibitory concentration (AUC/MIC) of 400–600 mg/L/hour on the basis of two or more measurements [76], while the more traditional approach relied on target trough levels as a surrogate marker [77]. 

Teicoplanin was discovered a few decades later [78], but with a similar mode of action, with clinically relevant differences pertaining to the MIC variability in some CoNS. In a French study, as much as one third of CoNS isolates were found to have an MIC of >4 µg/mL, and thus were non-susceptible to teicoplanin [79].

### 4.3. Lipoglycopeptides

Telavancin, dalbavancin, and oritavancin belong to the newly developed semisynthetic group of glycopeptides. Hence, the spectrum of antimicrobial activity resembles those of vancomycin and teicoplanin [80,81,82]. Their main indications per approval are acute bacterial skin and skin structure infections; in addition, telavancin can be used in pneumonia caused by MRSA as well. Intriguingly, dalbavancin offers the possibility of once-a-week dosing, owing to its long half-life of 8.5 days. Side effects of note are renal impairment (telavancin); gastrointestinal symptoms and liver enzyme elevation (dalbavancin); and infusion reaction and potential drug–drug interactions (oritavancin). Experience from extended clinical use is still relatively limited.

### 4.4. Daptomycin

Daptomycin is a cyclic lipopeptide that was discovered in the 1980s; however, its approval in the US and in Europe followed decades later in 2003 and 2006, respectively. This substance targets the cell membrane of Gram-positive bacteria. Its spectrum of activity is similar to that of glycopeptides. A low rate of resistance among CoNS has recently been reported [83]. Examples for its clinical use include prosthetic joint infections by oxacillin-resistant staphylococci with either non-susceptibility or reported allergy to vancomycin [84] and bacteremia by *S*. *aureus*, including right-sided endocarditis. Initially, this substance was approved primarily for acute bacterial skin and skin structure infections caused by Gram-positive cocci. It has also been proposed as an alternative to linezolid in the treatment of VRE infections, albeit with a higher than approved dose, i.e., 8–12 mg/kg per day [85]. Relevant adverse effects are a reversible muscle toxicity, which was especially prevalent during the early years when daptomycin was given twice daily, and eosinophilic pneumonia [86]. 

### 4.5. Oxazolidinones

Linezolid is the most widely used oxazolidinone, a group of bacteriostatic drugs that inhibit protein synthesis. The main role of linezolid has been as a therapeutic option for VRE infections. Currently, only limited data is available to recommend its use in CoNS as a first-line treatment. Still, there have been increasing reports on linezolid use in the past years [87,88]. In a large study on the feasibility of early oral step-down therapy in endocarditis, linezolid was among the antibiotics that were used, and was largely used in combination with other antibiotics [89]. The study which rightfully gained a lot of attention by showing the non-inferiority of early oral step-down therapy to the conventional intravenous therapy, was however underpowered to assess individual antibiotic combinations and their comparison among each other. Nevertheless, the continuing trend to shorten intravenous therapies in favor of an early step-down is likely to keep linezolid relevant, due to its oral formulation with high bioavailability. However, imprudent use of linezolid has been linked to the emergence of linezolid-resistant strains [90], in particular linezolid-resistant *S*. *epidermidis* [91,92,93]. CoNS have a higher and easier ability to acquire and develop linezolid-resistance factors following exposure to the drug. The incidence of linezolid resistance in CoNS is currently higher than in *S*. *aureus*. It is believed that linezolid resistance originally emerged in CoNS, and then transmitted to *S*. *aureus* [94]. Another concern is severe and partially irreversible side effects, such as bone marrow toxicity, lactic acidosis, and neuropathies. The risk for adverse events steeply increases after 2 weeks. Another limitation is the restriction of approval for a maximum of 28 days, which hampers its longer-term use in, e.g., bone and joint infections [95].

### 4.6. Alternatives and Biofilm-Active Substances

Alternative substances that are of importance, especially in non-critical, localized infections, such as skin and soft tissue infections or bone and joint infections, are e.g., co-trimoxazole, clindamycin, and doxycycline. All of these possess a high oral bioavailability [96]. In foreign body-related infections, the additive use of a biofilm-active substance has been propagated by relevant guidelines and experts [84,97,98]. For CoNS, these are largely rifampicin and, to a lesser extent, fosfomycin [99]. 

## 5. Conclusions

In conclusion, CoNS form a large group of skin microbiota that play an increasingly important role, especially in health-care associated infections. The advent of MALDI-TOF MS in routine microbiology diagnostics and the increasing feasibility of whole-genome sequencing will bring even more CoNS species to life, while also helping to better characterize their pathogenicity and resistance profiles. The main challenge in routine diagnostics remains to correctly assign the causative role of CoNS recovered from primarily sterile materials, as no single or composite diagnostic tool with a sufficiently high sensitivity and specificity exists to allow for a reliable diagnosis. Future studies aiming to leverage both modern pathogen detection methods and assessment of the host response could help shed light on the clinical utility on such a combined approach. Moreover, clinical data are highly needed especially for the rarer and newly described CoNS.

## Figures and Tables

**Figure 1 microorganisms-09-00830-f001:**
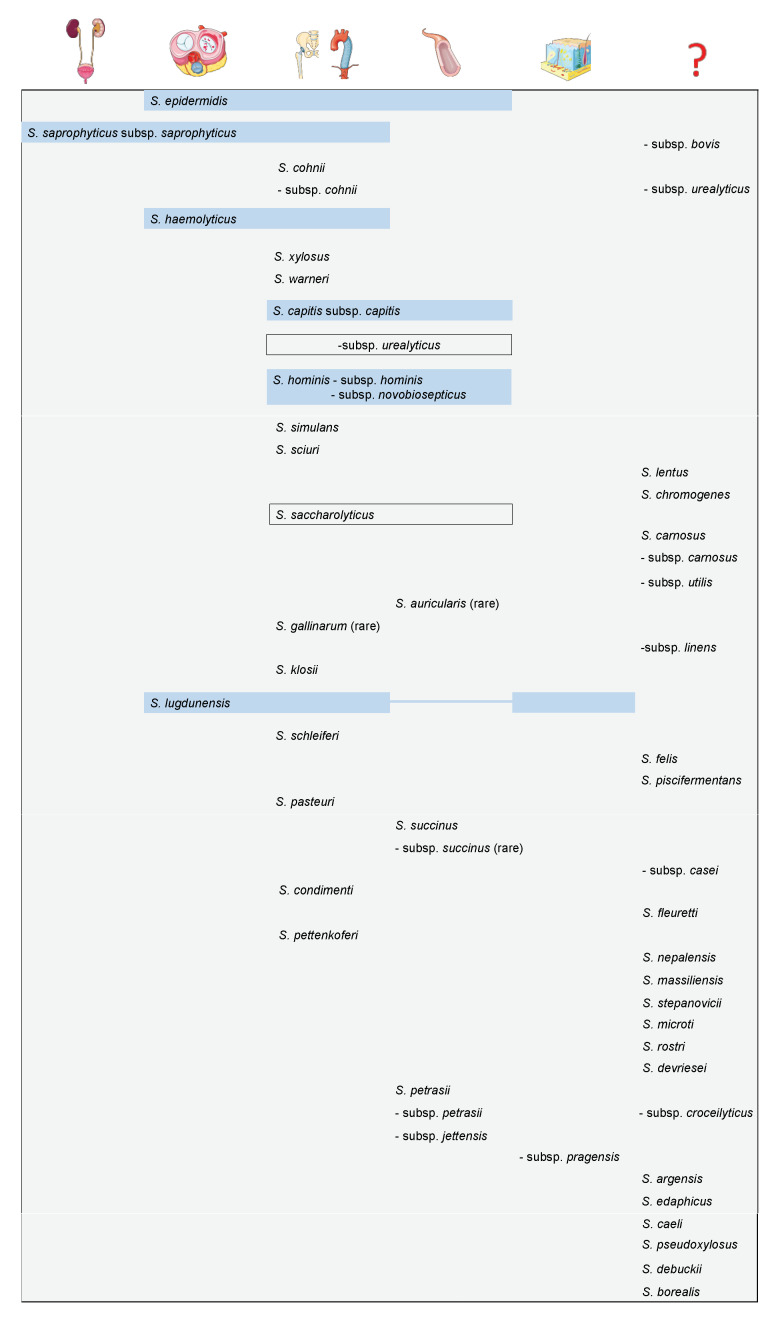
Overview of CoNS, according to the predominantly affected organ site or infectious syndrome, from left to right: urinary tract; cardiac valves; prosthetic joints and vascular grafts; bloodstream infections; skin and soft tissue infections; the question mark designates species of unknown clinical relevance. Most prevalent and relevant species are highlighted in blue; species are depicted in accordance with their first description, from top to bottom. Pictograms were taken with permission from Servier Medical Art [21].

**Figure 2 microorganisms-09-00830-f002:**
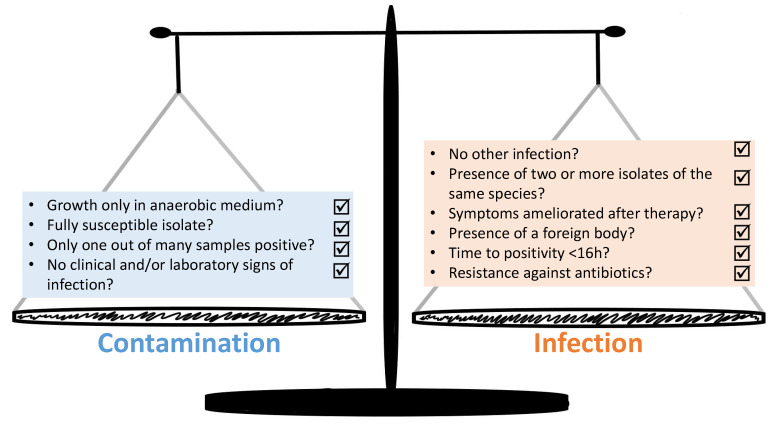
Model for differentiation between contamination and infection pertaining to the finding of CoNS in a primary sterile material; BC: blood culture.

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
