# Peer review of "Update on Coagulase-Negative Staphylococci—What the Clinician Should Know"

_microorganisms, 2021, doi:10.3390/microorganisms9040830_

Round 1

Reviewer 1 Report

In this review manuscript, the authors describe the range of variation among clinical isolates of coagulase-negative staphylococci (CoNS), attributes of CoNS infections, host attributes associated with CoNS infection risk, challenges associated with identifying CoNS infections, and considerations for CoNS therapy options.

General comments

1. The figures add little to the paper and in some cases are not well explained. For example, Figure 1 is fully redundant with the text and does little more than list the characteristics discussed. There is a similar issue with Figure 3. With Figure 2 the meaning of the red lightning bolts is not stated in the figure legend. Similarly, some aspects of Figure 2 are difficult to interpret. For example, which organ site or infectious syndrome is indicated for S. saccharolyticus? Also, rather than having to refer to the figure legend to interpret the organ site or infectious syndrome column headings, I think it would be easier to simply use text headings. I do not understand the meaning of the “?” heading—perhaps “unknown clinical relevance” should be “known clinical relevance?

2. There are several instances of awkward phrasing in this manuscript that should be revised for clarity of meaning. For example, the sentences on lines 40-45 could be simplified and clarified as follows: “Many CoNS infections are associated with foreign bodies (e.g. catheters) that facilitate CoNS biofilm formation, which contributes to CoNS pathogenicity. Importantly, the virulence factors of CoNS vary considerably, which some species (e.g. S. lugdunensis) being capable of significant adverse impacts on patients.“

An important aspect of this revision is the fact that virulence is the outcome of an interaction, not an attribute that can be possessed.

Specific comments

  1. Line 22. I suggest changing “substances” to “drugs.”
  2. Lines 42. Please elaborate on what these “prominent properties” include.
  3. Line 46. “…also non-pathogenic commensals of the human skin.” Could the same be said of S. aureus or MRSA who can be commensal members of skin microbiota? Perhaps, revise to indicate that CoNS are common commensals that are seldom problematic?
  4. Line 80. Change “predestined” --> “at high risk”
  5. Lines 88 and 106. Source of images stated but not clear that authors have permission to use the images.
  6. Lines 89-92. I think this topic warrants additional discussion.
  7. Lines 92-93. Typo. “urrently” --> “Currently”
  8. Lines 118-119. Typo. Extra line break
  9. Lines 127-129. Why select these three species for this description? I think some context for this would be helpful.
  10. Lines 231-251. This discussion of vancomycin offers little direct tie to CoNS and so feels out of place in this review.
  11. Lines 231-251. Typo. Italics
  12. Line 283. “member of this group of” --> “oxazolidinone”

Reviewer 2 Report

The manuscript entitled “Update on coagulase-negative staphylococci – what the clinician should know” is a significant study in the present scenario. The role coagulase-negative staphylococci (CoNS) as causative agents of serious nosocomial infections in humans is increasing rapidly. Although CoNS infections are less severe than Staphylococcus (S.) aureus infections, their treatment is more complicated because of the dramatic increase in antibiotic resistance.

I have read with great attention and interest the topics of the manuscript, which is definitely impressive.  

The major problem in the manuscript is the difficulty of the language which is not feasible to read and understand. Many sentences or even paragraphs are misread and give more than one meaning or intention. I think the language was translated directly from German to English and I do believe that the most of the manuscript needs to be rephrased before being available for publishing.

I have other comments, which are listed below.

Abstract:

Line 22: second-line substances: This needs to be more clear. Do the authors mean antimicrobial drugs?

Introduction:

Line 29: Please mention the recent spp. That has been described in 2020

Line 33: The term “staphylococci other than Staphylococcus aureus” (SOSA) should not be used instead of the designation "CoNS" as SOSA group contains other than S. aureus-coagulase positive group members also.

Lines 34, 35: Please delete this sentence (A PubMed/MEDLINE search for “Staphylococcus aureus” performed in February 2021 re- 34

trieved 126,429 findings).

Lines 40, 41: Please provide a reference for this information: In many cases, the infection is related to the presence of foreign bodies, which may be favored by the ability of CoNS to form biofilms.

Line 47: rather than the causative agent of infection.

Increasing Clinical Impact of CoNS:

Lines 53-55: Please rephrase.

Lines 67, 68: CoNS represented 31% among all cases of nososcomial BSI within a period of seven years in a total of 49 US hospitals.

Line 73: More recent studies revealed the relationship between CoNS and the increased application of implanted medical devices like cardiac valves or joint replacements.

Lines 76 and 77: It is better not to form a question.

Line 78: Please use (surface) instead of (indwelling), as a result, to enable….

Line 79: evade the host immune system or antibiotics: These are totally different 2 aspects. Biofilm is a virulence factor that enables the microorganisms to protect itself against the action of antibiotics not to evade them.

Line 82: Please use (Additionally) instead of And even…

Line 83: As such, S. saprophyticus is a main cause for urinary tract infections….

Line 84: Figure 1 provides an overview about the factors which favor an infec- 84

tion with CoNS. Factors contribute CoNS-related infections are illustrated in Figure. 1.

Figure. 1: Patient characteristics and groups that confer risk for clinically relevant CoNS- infections.

Line 93: Currently, more than 50 different CoNS species have been described Ref??

Line 95: the associated infectious syndrome they are associated with. In total, six species are widely accepted believed to be associated with a higher clinical significance.

Lines 98-100: Antibiotic resistance has become an increasingly probleme with S. lugdunensis, which has some mutual features with S. aureus.

Please illustrate some of the infections caused by S. lugdunensis with related references.

Please mention in short brief with some examples for the role of animals in transmission of zoonotic CoNS and associated resistance and virulence mechanisms. 

Line 102: Overview of coagulase-negative staphylococci CoNS.

Line 103: It is better to give numbers instead of from left to right: urinary tract; cardiac valves….

Line 106: The reference (23) is just the website. Please specify the exact link at which all the species and subspecies with their predilection sites are illustrated.

Line 111: Like S. argensis, S. edaphicus was also isolated….

Lines115 and 116: S. edaphicus a resistance reservoir and an intermediate for that potentially spreads resistance associated these genes.

Lines 116 and 117: All references (27, 28 and 29) should be illustrated together at the end of the text.

Lines 124: impacts on humans beings due to the lack of data. Thus, further studies and case reports will help to better understand their role and potential risk to humans.

Lines 131 and 132: Only one case report can be found for S. massiliensis, S. petrasii subsp. petrasii, and S. petrasii subsp. croceilyticus each is existing.

Line 133: translated

Lines 134-137: Do the authors mean in this paragraph that many CoNS species are isolated together from human samples with difficulty to identify the main causative agent of pathogenesis?

  1. The Diagnostic Complexitis – How to Distinguish between Infection and Contamination

Line 141: CoNS-member

Line 148: please add factor instead of layer

Line 150: It would be better to talk about pandemics in general not to specify COVID-19

Line 165: Not only do appearances differ: Do the authors mean characteristics?

Line 173: please remove explained in the following.

Figure 3: Growth only in anaerobic medium? They can also grow in aerobic conditions.

Lines 177 to 189: The whole paragraph needs to be rephrased.

Table 1: almost always in mixtures with other CoNS,….

  1. Therapy and Prevention – many old bugs, little new drugs

Please state the main antibiotic resistance traits in CoNS for the mostly used antimicrobial drugs including penicillin, oxacillin/methicillin, gentamicin, clindamycin, ciprofloxacin, and erythromycin. Please also add a paragraph about the ability of CoNS of rapid acquisition, possessing, and modification of resistance genes. This ability subsequently promotes the transmission of these genes into different staphylococcal species or even other bacterial genera (1-3).

  1. Becker K., Heilmann C., Peters G. Coagulase-negative staphylococci. Clin. Microbiol. Rev. 2014;27:870–926. doi: 10.1128/cmr.00109-13.
  2. Otto M. Coagulase-negative staphylococci as reservoirs of genes facilitating MRSA infection: Staphylococcal commensal species such as Staphylococcus epidermidis are being recognized as important sources of genes promoting MRSA colonization and virulence. Bioessays. 2013;35:4–11. doi: 10.1002/bies.201200112
  3. Moawad AA, Hotzel H, Awad O, Roesler U, Hafez HM, Tomaso H, Neubauer H, El-Adawy H. Evolution of Antibiotic Resistance of Coagulase-Negative Staphylococci Isolated from Healthy Turkeys in Egypt: First Report of Linezolid Resistance. Microorganisms. 2019 Oct 22;7(10):476. doi: 10.3390/microorganisms7100476. PMID: 31652567; PMCID: PMC6843140.

Lines 231-251:  Please correct the italic writing

Line 231: has been considered as a reliable drug since its introduction in….

Line 232: the 1950s with a broad Gram-positive coverage throughwhich it exerts by inhibiting cell wall synthesis

Line 259: Telavancin, dalbavancin, and oritavancin belong to the newly developed semisynthetic group of glycopeptides.

Line 271: targets

Line 273: reported to be low

Line 282: Please add a paragraph stating that CoNS have a higher and easier ability to acquire and develop linezolid-resistance factors following exposure to the drug. The incidence of linezolid resistance in CoNS is currently higher than S. aureus (4). It is believed that linezolid resistance originally emerged in CoNS then it was transmitted to S. aureus.

  1. Gu B., Kelesidis T., Tsiodras S., Hindler J., Humphries R.M. The emerging problem of linezolid-resistant Staphylococcus. J. Antimicrob. Chemother. 2013;68:4–11. doi: 10.1093/jac/dks354.

Line 286: there have been increasing reports on Linezolid use in the past years.

Line 297: Please provide a reference stating these adverse effects.

The genotypic resistance associated markers were not taken into account through the whole review. I ask the authors to add more information about genetic resistance associated markers related to the illustrated phenotypic resistance profiles.

References

Two main issues, the authors should take in consideration please:

  1. Please provide the DOI for all references.
  2. The huge number of references has to be reduced.

To sum it up, I do believe that the manuscript needs to be improved to be suitable for publishing. Major revision is needed for the manuscript

Round 2

Reviewer 1 Report

In this review manuscript, the authors describe the range of variation among clinical isolates of coagulase-negative staphylococci (CoNS), attributes of CoNS infections, host attributes associated with CoNS infection risk, challenges associated with identifying CoNS infections, and considerations for CoNS therapy options. I found the clarity and readability of the revised version considerably improve and I think that overall the authors have responded well to the comments provided by me and the other reviewer.

Specific comments

  1. I have not issue with the authors wanting to retain Figures 1 and 2 (formerly Figures 2 and 3). However, I am still unable to interpret organ site or infectious syndrome of S. saccarolyticus in Figure 1. There is a similar issue with S. capitis subsp. capitis subsp. urealyticus. If I’m understanding correctly each of these are associated with both “joint prothesis and vascular grafts” and bloodstream infections. If so, there are a few ways this could be more clearly indicated—maybe a box about the corresponding cells?
  2. Suggested revision. Line 65. “application” --> “use”
  3. Suggested revision. Line 74. Remove “also”
  4. Suggested revision. Line 256. “non-susceptibility” --> “resistance”

Author Response

Specific comments

  1. I have not issue with the authors wanting to retain Figures 1 and 2 (formerly Figures 2 and 3). However, I am still unable to interpret organ site or infectious syndrome of S. saccarolyticus in Figure 1. There is a similar issue with S. capitis subsp. capitis subsp. urealyticus. If I’m understanding correctly each of these are associated with both “joint prothesis and vascular grafts” and bloodstream infections. If so, there are a few ways this could be more clearly indicated—maybe a box about the corresponding cells?

Response: We thank Reviewer #1 for their opinion on Figure 1. We have complied with the Reviewer’s suggestion and have provided boxes around the respective cells.

  1. Suggested revision. Line 65. “application” --> “use”
  2. Suggested revision. Line 74. Remove “also”
  3. Suggested revision. Line 256. “non-susceptibility” --> “resistance”

Response: We thank Reviewer #1 for these suggestions. We have changed the respective paragraphs accordingly. Many thanks to both Reviewers and the Editors for their help in substantially improving the manuscript.

Reviewer 2 Report

Hereby, I accept the current version of the manuscript to be considered for publication.

I would like to thank the Editors and Authors

Author Response

Hereby, I accept the current version of the manuscript to be considered for publication.

I would like to thank the Editors and Authors.

Response: We thank Reviewer #2 for their help in improving the manuscript.